# Benefit–cost analysis of electronic claims processing under Ghana's National Health Insurance Scheme

Justice Nonvignon ![ORCID] ,[1] Rebecca Addo,[2] Huihui Wang,[3] Anthony Seddoh[4]

¹Department of Health Policy, Planning and Management, University of Ghana School of Public Health, Accra, Greater Accra, Ghana
²Centre for Health Economics Research and Evaluation (CHERE), University of Technology Sydney, Sydney, New South Wales, Australia
³Health, Nutrition and Population Global Practice, World Bank Group, Washington, DC, USA
⁴Health, Nutrition and Population Global Practice, World Bank Group, Accra, Ghana

**Correspondence to**
Justice Nonvignon;
jnonvignon@ug.edu.gh

## ABSTRACT

**Objective** The aim of this study was to evaluate the benefit–cost of E-claims. A benefit–cost analysis was used to evaluate the efficiency of E-claims from the perspective of the providers and the purchaser.

**Design** A benefit–cost analysis approach was taken for this economic evaluation. Furthermore, we estimated the incremental benefit–cost ratio (IBCR) of the intervention under assessment.

**Participants** Purchasers and healthcare providers of the National Health Insurance Scheme (NHIS) of Ghana were the study population.

**Results** The analysis was stratified according to providers and purchaser. Cost incurred in processing claims electronically and manually were estimated by assessing the resource use and their corresponding costs. Sensitivity analysis was conducted to assess the robustness of the results to variations in discount rate and proportions of claims processed under E-claims compared with paper claims. The combined sample of providers and purchaser made incremental gains from processing claims electronically. The IBCR was −19.75, 25.56 and 5.10 for all (sample) providers, purchaser and both providers and purchaser, respectively. When projected for the 330 facilities submitting claims to the NHIS claims processing centre (CPC) as at December 2014, the IBCR were −35.20, 25.56 and 90.06 for all providers, purchaser and both providers and purchaser. The results were sensitive to the discount rate used and proportions of E-claims compared with paper claims.

**Conclusion** Electronic processing of claims is more efficient compared with manual processing, hence provide an economic case for scaling it up to cover many more healthcare facilities and NHIS CPCs in the Ghana.

## BACKGROUND

Prior to the establishment of Ghana's National Health Insurance Scheme (NHIS) in 2003, the health financing system in Ghana was characterised by high out-of-pocket payment at the point of service, low private health insurance penetration and few community-based health insurance schemes. Previous studies have provided a comprehensive historical overview of health financing in Ghana,[1] and other contexts of the NHIS.[2–4] Other studies explored PPMs and sustainability of expenditure under the NHIS.[5–11]

## STRENGTHS AND LIMITATIONS OF THIS STUDY

⇒ The study provides evidence for scaling up of the E-claims system which improves the efficiency of the National Health Insurance Scheme compared with the manual processing of claims.
⇒ This study addressed important gaps in knowledge about electronic processing and manual processing of insurance claims in a resource-constraint setting.
⇒ There are difficulties in quantifying some of the benefits of the E-claims into monetary terms, reflecting methodological challenges with cost–benefit analysis.
⇒ There is little evidence available regarding the use of electronic claims processing that provides a comparison to Ghana, therefore, it was difficult to identify similar studies

The main objective of the NHIS as contained in the Act 650 establishing it (and later Act 852 2012) was 'to secure the provision of basic healthcare services to persons resident in the country'.[12] The NHIS is mainly financed by value-added tax and Social Security National Insurance Trust deductions, covering 95% of the disease conditions afflicting the population. A Free Maternal and Child Care policy was also introduced in 2008 with support from the British Government[8] to accelerate attainment of MDG 4 and 5, under which antenatal, delivery and postnatal care services to pregnant women were made free to clients.[5] Faced with a number of issues around governance and operations, legalities and definition of exempt groups, the National Health Insurance Act (NHIA), 2012 (Act 852) was passed in 2012[12] replacing Act 650. By the end of 2014, the NHIS had a total active membership of 10 545 428 representing 39% of the total population of Ghana.[13]

Since its implementation in 2004, the NHIA, the regulator of the NHIS, has used various provider payment mechanisms (PPMs) to reimburse health service providers for the use of services by NHIS members. At the start of the implementation in 2004,

itemised billing with no standard fee schedule was used to reimburse providers for services and medicines. Under this PPM, providers negotiated reimbursement rates with the NHIS office at the district level. In 2008, the Ghana-Diagnostic Related Grouping (G-DRG) payment mechanism was introduced to address challenges such as the inability of healthcare providers to code claims properly (resulting in the submission of incomplete claims); escalating costs due to spurious claims and the lack of a system to monitor fraud. The G-DRG was used to pay for services and procedures while payment for medicines was made using standardised itemised fees (fee-for-service) based on a medicines list.[5 7] Under the G-DRG payment system, facilities submit claim forms filled for each outpatient visit and inpatient episode (coded under G-DRG) to NHIS for reimbursement. In addition, due to continued cost escalation, some of which has been attributed to fraud and moral hazards,[14] per capita (capitation) payment mechanism for primary healthcare services was introduced in 2012. This was piloted in the Ashanti region[5 7] and was scaled up to two additional regions but was halted in 2017.

In the search of a more efficient method of reimbursing providers while financially sustaining the NHIS, the electronic claims (E-claims) processing was introduced in 2013.[11] This initiative was part of the World Bank-supported health insurance project (HIP) whose main objective was to create an enabling environment for the scheme and facilitate its financial and operations management. The E-claims processing was piloted in 29 health facilities (providers) by the end of the project in March 2014. In 2019, the E-claims covered 120 healthcare facilities across the country and received a quarter of total volumes of claims submitted to NHIA, the purchaser.

### Description of E-claims system

E-claims refers to any claim that is captured on a CD, Memory stick, online real time, through a Web Front End model or through Health Management Information System. The purpose of introducing electronic claims was to minimise the extent of human interference in the claims adjudication process, minimise errors association in claims processing from the healthcare provider end and its adjudication process at the NHIA. At the insurance side, a national claims processing centre (CPC) was equipped to process electronic claims from providers across the country. At the service provider side, 29 health facilities were initially selected to submit claims through two options: (1) an XML (Extensible Markup Language) interface that transmits claim information from the provider's HMIS and electronic health record system, and (2) web-based tool for entering and submitting claim information. These claims are submitted to the purchaser, the NHIA' CPC in Accra, where they are equipped to electronically vet and process the E-claims submitted by providers. Under paper-claims system, on the other hand, providers fill out claims forms per patient and submit all the forms compiled over a period to the purchaser for reimbursement.

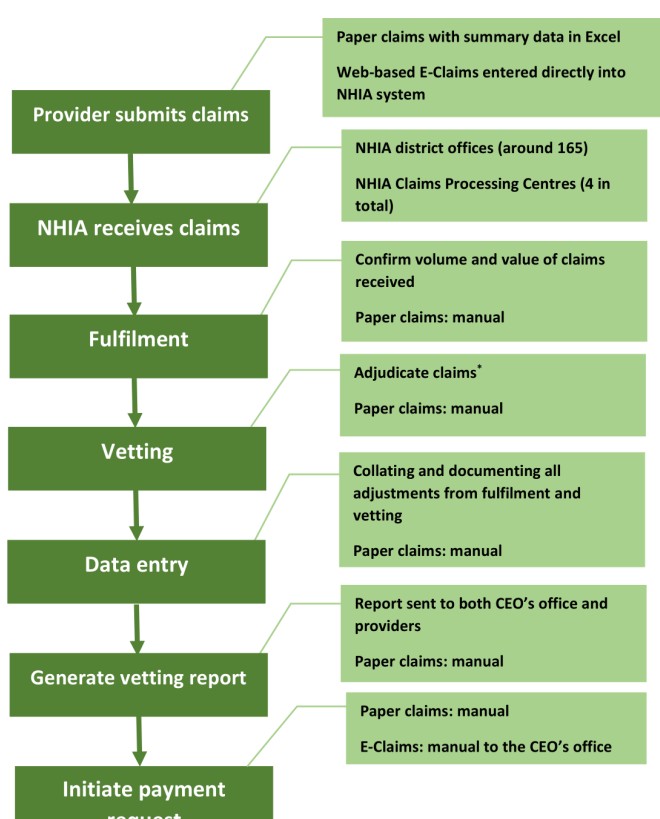

**Figure 1** Claims processing flow chart at the National health Insurance Authority. Source: Authors' construct based on interviews with NHIA staff. NHIA, National Health Insurance Act

In terms of claims processing, both E-claims and paper-claims systems follow the same steps to process claims for reimbursement. For instance, at the purchaser side, once the CPC receives claims from the providers, they confirm the volume and value of the claims received. Afterwards, the claims are vetted to assess providers' adherence to eligibility, benefit package, Ministry of Health (MOH) treatment protocols, diagnosis, prescribing levels, treatment and tariffs assigned to each claim. Any deviation from the NHIS requirements are accounted for by adjusting claims submitted, after which reports are written and submitted to the providers and the Chief Executive Officer of the NHIA. The difference between the electronic and manual claims is that whereas the processing steps outlined above are automated either fully or partially under the former, the later are carried out manually. At the provider's side, once they submit the claims to the CPC either electronically or by claim forms, data are entered for the manual claims, but the remaining steps carried out by the CPC as described above. Figure 1 presents the steps involved in processing claims under the NHIS. Both processes require labour capacity to undertake entry of data into the web-interface for the E-claims and filling and compiling of forms under the manual claims. In addition to this, a provider requires a computer, UPS, sustained internet connection, staff with the appropriate skills and electricity

to submit E-claims. Details of NHIA claim review and reimbursement process can be found elsewhere.[9]

At the end of the pilot phase, a number of benefits were observed for the service providers, scheme managers and in the interactions between providers and scheme managers. For service providers, there were (1) cost savings with regards to operations cost by not using paper; (2) decreased rejection rate due to better quality claims; (3) decreased turn-around time for reimbursement and (4) increased fund stability. For scheme managers, there were (1) saved operation and management cost for claim processing by not using cumbersome paper-based system; (2) saved operation and management cost for clinical auditing; (3) increased capacity and efficiency to identify fraud claims; (4) increased capacity and efficiency to identify inappropriate services and improve service quality and (5) increased capacity to provide evidence for policy-making and implementation. Finally, with regard to the benefits accrued in the interactions between providers and scheme managers, there were, (1) increased capacity to link up with digitalised membership management system, electronic insurance payment system, electronic medical records system in hospitals and health information management system and (2) improved human resource capacity to implement administration requirement of the health insurance scheme.

To demonstrate the effectiveness of E-claims over paper claims processing, Nsiah-Boateng *et al*,[9] used a cross-sectional study to compare the ability of E-claims and paper claims processing to detect spurious claims and reduce costs to the NHIA. The authors found that the E-claims review system had the ability to reduce cost to the NHIA more than paper claims (17% vs 4.9%). While this study demonstrates that E-claims is effective in reducing cost to the NHIA compared with paper claims processing, the costs and benefits of processing claims electronically or manually to providers and the overall health system is not known. A comprehensive economic evaluation could contribute to informed decision on which claims processing type provides value-for-money to both providers and the overall health system. Therefore, the aim of this study was to assess the costs and benefits of E-claims compared with the paper claims processing to inform policy-makers, as well as fill a gap in the literature.

## METHODS
### Study design
A benefit–cost analysis approach was taken for this economic evaluation. BCA assesses the costs and benefits of an intervention compared with the next best alternative. Unlike other types of evaluations, Benefit CostAnalysis assesses the monetary value of the benefits.[15–18] Its theoretical foundations are from the concept of Pareto efficiency, where an additional reallocation is acceptable only if it makes at least one person better off without making another person worse off.[19] In this study, we estimate the incremental benefit–cost ratio (IBCR) of

the intervention under assessment. The IBCR in a BCA denotes the net benefit of an intervention compared with its best alternative. Therefore, using BCA, an intervention is said to be efficient compared with the next best alternative if the net benefit is positive.

Thus, this study sought to assess the efficiency of the E-claims compared with paper-claims system, and to establish if the marginal benefits of processing claims electronically at least covers its marginal cost: implying positive net benefits, or not. We report our methods and results in accordance with the Consolidated Health Economic Evaluation Reporting Standards.[20]

### Study population
Purchasers and healthcare providers of the NHIS of Ghana were the study population. As at December 2014, there were 330 district hospitals, eight secondary hospitals and one tertiary hospital submitting claims, out of which 29 were implementing E-claims. The purchaser selected was the CPC of the NHIS. For the purposes of this evaluation, 11 providers were purposively sampled out of a total of 29 providers implementing E-claims; four districts and five regional hospitals who were processing their claims electronically were selected based on the start dates of the E-claims system (ie, we selected facilities that had processed E-claims for at least 1 year prior to evaluation); the police hospital and the 37 military hospital were also selected based on their experience with implementing the E- claims. Therefore, in addition to estimating the overall net benefit of E-claims processing in Ghana, the analysis was further categorised into the following subgroups; district, regional, tertiary and central processing centre. No patient level data were accessed as part of this study.

### Perspective of analysis
The evaluation was conducted from the perspective of the Ghana health system to establish whether it was worthwhile for the system to invest in the E-claims by extending its coverage to all providers and NHIS processing centres.

### Estimating costs
Data on resource use and their associated costs were obtained from the NHIA and providers under study. At the providers' side, annual cost data were collected—year 2012 for costs involved in processing paper claims (all facilities under study were using paper claims) and year 2014 for costs involved in E-claims processing (all facilities under study had rolled out E-claims processing fully). At the purchaser side, data were collected for only year 2014 for both the costs of processing paper claims and E-claims. This was because at the time, the CPC was processing both types of claims since not all providers had been enrolled onto the E-claims system. However, the cost data collected were disaggregated into paper and electronic claims by weighting the total claims by the proportions of E-claims and paper claims processed during 2014.

The costs information collected included capital and recurrent costs. Capital costs included the costs of

installation (server and software purchase/upgrade, internet) and equipment (printers, computers, air conditioner, routers and inverter). Recurrent costs included labour, transport, staff training, operations costs (office space/storage space rental costs, furniture) electricity bills, and stationery.

The total capital costs incurred by the providers were calculated by summing the costs attributable to each capital cost item. Furthermore, since the start-up costs for the E-claims reflect costs of inputs spanning over more than 1 year, the equivalent annual cost (EAC) of capital items was derived by dividing the total capital cost (summed for all units of a given capital item) by the annualisation factor. This was done to enable summation of capital costs with recurrent costs that occurred within 1 year. The annualisation factor was derived from annuity tables using the useful life of each capital item (3 years were assigned for printers, UPS, stabiliser, router and batteries and 5 years were assigned for other capital items including computer based on discussions with providers) and a discount rate of 3%, as recommended for the conduct of economic evaluation in low middle income countries.[21–23] Thus, the total capital cost (for E-claims and paper claims) for a provider was derived by summing the EACs for all capital inputs.

The total cost incurred by the provider for each recurrent item was also calculated by multiplying the unit costs by quantity. Total recurrent cost was calculated by summing the recurrent costs that providers attributed to recurrent items they used in processing both E-claims and paper claims. Data on the number of staff under each staff category involved in processing claims (E-claims and paper claims) for each provider together with the proportion of time that each staff spends per month on claims processing and the gross monthly salary of each staff were collected. The labour cost per staff attributable to claim processing was calculated by multiplying the total monthly gross salary by the proportion of time spent. This was summed over 1 year and across staff categories to derive the total labour costs, which were then added to the recurrent costs. The costs of the paper claims were converted into 2014 estimates: base year of analysis, using the consumer price indices for health goods collected from the Ghana Statistical Service.[24] The total costs per providers were estimated by summing capital and recurrent costs. Cost per claim processed under each type of claims processing were derived by dividing the total costs by the total number of claims processed.

From the purchaser's side, the total operational costs (ie, excluding capital and other investment costs) of the NHIA for 2014 was collected and used to apportion operational costs to the CPC based on the proportion of staff in the CPC as compared with the total number of staff of NHIA. The assumption here is that the proportion of staff reflects the proportion of not only staff time but also other operational expenses that could be attributed to the activities of the CPC. This was done as it was difficult to disaggregate specific cost items for the CPC. Once

the operational costs attributable to CPC was derived, the proportion of E-claims or paper claims processed by the CPC was used to apportion the CPC operational costs to either E-claims or paper claims. Then, the cost per claim was derived by dividing the total costs of E-claims/paper claims by the total volume/number of such claims processed by the CPC. The cost analysis was performed in Ghana Cedis (GHS) and converted into US Dollars (US$) using the US$-GHS exchange rate for 30 June 2014 (3.00) accessed from www.xe.com. The choice of cost parameters was based on the relevant costs incurred under both claims processing types, given the types of resources expended on various activities undertaken as part of claims processing.

### Estimating benefits

The benefits of claims were assessed using: (1) volumes and values of claims reimbursed and (2) claims rejection rate. For the providers, the volumes of claims were the total number of claims submitted for reimbursement. Subsequently, the values of the claims were estimated as their expected payout from the NHIS. Claims rejection rate was estimated as the difference between the claims submitted for reimbursement and the actual number of claims that were reimbursed by the NHIS. In other words, the difference between the projected value claims submitted and the actual payout received from the NHIS. The claims rejection rate was estimated for each month and that for the 1-year period estimated as the average rejection rate per month for year 2012 and 2014 for paper claims and E-claims, respectively.

Therefore, benefits of E-claims and paper claims per provider was estimated as the difference between the value of the submitted claims and the value of claims reimbursed (accounting for claims rejection rate). The value per claim submitted per provider was calculated by dividing the total value of claims submitted by the total volume of claims submitted.

On the other hand, the benefits of either types of claims processing to the purchaser (NHIA CPC) was estimated as the costs saved from the payouts that were not made due to rejection of submitted claims from providers. Thus, they were calculated as the cost due to rejected claims; that is, the difference between the expected payout for claims submitted by providers and the actual payout made after adjusting for claims rejected. The choice of benefit parameters was based on realistic availability of data from both providers and the purchaser.

### Estimating incremental benefit–cost ratios

Incremental costs were calculated as a difference between the cost of E-claims and paper claims. The incremental benefits were calculated as the difference between the benefits of the E-claims and paper claims (table 1). The IBCRs, a measure of efficiency, were therefore estimated for each provider and CPC (or purchaser) by dividing incremental benefits ($B_{\text{E-claims}} - B_{\text{paper claims}}$) by incremental costs ($C_{\text{E-claims}} - C_{\text{paper claims}}$). The IBCR per specific unit (that

**Table 1** Detailed calculation formulas

| Estimating costs | | |
|---|---|---|
| **Provider** | | |
| | Total costs | Total recurrent costs+total capital costs |
| | Capital costs | Sum of equivalent annual cost of capital |
| | Equivalent annual capital cost | Current cost of capital/ annualisation factor |
| | Recurrent cost | Unit cost * quantity for each item |
| **Purchaser** | | |
| | Total operational cost of CPC | Total operational cost of NHIA * proportion of CPC staff to total staff |
| | Operational cost of E-claims or paper claims | Total CPC operational cost * proportion of staff processing e-claims or paper claims |
| **Estimating benefits** | | |
| **Provider** | | |
| | Benefits of E-claims or paper claims | Total value of submitted−total value of reimbursed claims |
| **Purchaser** | | |
| | Cost savings | Expected payouts from submitted claims −actual payout/reimbursement |
| **Incremental benefit–cost ratio** | | |
| | Incremental cost | Total cost of e-claims−total cost of paper claims |
| | Incremental benefit | Total benefit of e-claims−total benefit of paper claims |
| | IBCR | Incremental benefit/ incremental cost |

CPC, claims processing centre; IBCR, incremental benefit–cost ratios; NHIA, National Health Insurance Act.

is provider or purchaser) determines the efficiency of that unit. An IBCR greater than 1 indicates that the additional benefits of E-claims outweigh the additional costs. The IBCR was calculated for each provider, provider type (ie, district/regional/teaching hospital), purchaser and the entire health system (ie, both providers and purchaser). The analysis for provider type used the average costs and benefits of each provider type (ie, summing for all units in the specific provider type within the study sample and dividing by the number of units). The analysis for the health system was done by adding the costs and benefits of all providers and purchaser and estimating their IBCR.

Furthermore, the analysis was extended to cover total number of providers (by type) projected to submit claims to the CPC as at December 2014. It is worthy to note that 91% of these facilities were not processing their claims electronically at the time of data collection. The costs and benefits were calculated by multiplying the average costs and benefits of providers (according to type) to the total number of providers using E-claims at the time of evaluation. There was only one E-CPC for the NHIA as at 2014.

### Sensitivity analysis
Some key parameters of the study were varied to ascertain the robustness of the IBCRs estimates. A univariate sensitivity analysis was conducted on the discount rate and the proportions of claims processed under E-claims compared with paper claims. The discount rates were varied within a range of 0%–10%, excluding 3%, the base case.

### Patient and public involvement
Patients or the public were not involved in the design, or conduct, or reporting or dissemination plans of our research.

## RESULTS
### Costs of processing E-claims and paper claims
Table 2 presents the costs per claim for the two processing types for the study population: providers and purchaser. The average cost per E-claim for the providers was US\$0.65, US\$0.93 and US\$1.95 for the district, regional and tertiary hospitals, respectively. The providers also incurred an average cost per paper claim of US\$0.30, US\$1.16 and US\$2.26 for the district, regional and tertiary hospitals, respectively. The total cost per claim for all providers was US\$10.18 and US\$10.45 for E-claims and paper claims, respectively. Among the providers, the cost of processing (the cost of processing claims includes all costs incurred from preparing and submitting claims) both electronic and paper claims were highest among tertiary hospitals. In addition, Volta regional hospital spent the highest cost in processing E-claims followed by La General Hospital. Conversely, Atebubu hospital incurred the least cost in processing E-claims. In processing paper claims, Volta regional hospital incurred the highest cost followed 37 Military hospital. Takoradi hospital spent the least in processing paper claims. The purchaser; NHIS CPC incurred US\$0.59 for processing each E-claim and US\$0.50 for processing each paper claim at the CPC.

Recurrent costs accounted for a higher percentage of the costs of processing both E-claims and paper claims for providers and purchasers alike. For example, for district hospitals recurrent costs constituted 86% of E-claims processing costs and 96% of paper claims processing costs (see additional file for detailed description). The main driver of the recurrent cost was labour, followed by maintenance. It is worthy to note that start-up cost of E-claims contributed to 13.6%, 8.7%, 8.3% of the overall costs for district, regional and tertiary hospitals, respectively. Cost of labour also accounted for 50%–76% of costs of E-claims and 59%–83% of costs of paper claims.

The incremental cost for all providers was estimated at US\$−0.27 and that for the purchaser (CPC of the NHIA)

 

**Table 2** Estimated costs of processing E-claims and paper claims by healthcare providers and purchaser, 2014

| Study population | Cost per claim (US$) | | |
|---|---|---|---|
| | E-claims | Paper claims | Incremental cost (US$) |
| Providers | | | |
| District | | | |
| Atebubu hospital | 0.27 | 0.22 | 0.05 |
| St. Martins hospital | 0.59 | 0.29 | 0.3 |
| Takoradi hospital | 0.29 | 0.12 | 0.17 |
| La general hospital | 1.46 | 0.57 | 0.89 |
| *Average district* | *0.65* | *0.30* | *0.35* |
| *Total district* | *2.61* | *1.2* | *1.41* |
| Regional | | | |
| Sunyani hospital | 0.90 | 0.29 | 0.61 |
| Koforidua hospital | 0.61 | 1.25 | (0.64) |
| Ridge hospital | 0.81 | 0.32 | 0.49 |
| Police hospital | 0.74 | 1.95 | (1.21) |
| Volta hospital | 2.00 | 2.60 | (0.60) |
| Effia Nkwanta hospital | 0.56 | 0.58 | (0.02) |
| *Average region* | *0.94* | *1.17* | *(0.23)* |
| *Total region* | *5.62* | *6.99* | *(1.37)* |
| Tertiary | | | |
| 37 military hospital | 1.95 | 2.26 | (0.31) |
| All providers | | | |
| Average | 0.93 | 0.95 | (0.02) |
| Total | 10.18 | 10.45 | (0.27) |
| Purchaser | | | |
| CPC | 0.59 | 0.50 | 0.09 |
| Health system (all providers and purchaser) | | | |
| Average | 0.90 | 0.91 | (0.01) |
| Total | 10.77 | 10.95 | (0.18) |

All costs were estimated in 2014. A conversion rate of GHC3.00 per US$1 was used.
CPC, claims processing centre; E-claims, electronic claims.

was US$0.09. When stratified by types of providers, the average incremental cost per E-claim per district, regional and tertiary hospitals were US$0.35, US$−0.23 and US$−0.31, respectively.

### Benefits of processing E-claims and paper claims
Table 3 presents the benefits of E-claims and paper claims to the healthcare providers and the purchaser (NHIA). The average value per E-claim without errors (ie, expected payout) for the providers was US$9.29, US$14.96 and US$114.31 for the district, regional and tertiary hospitals, respectively. The average value per paper claim without errors was US$8.22, US$13.87 and US$18.26 for

the district, regional and tertiary hospitals, respectively. When put together, all providers were expected to gain US$128.31 per E-claim and US$125.52 per paper claim as claims payout.

The NHIA expected payout for processing claims electronically and manually were US$11.26 and US$9.71 per claim, respectively. However, after correcting for errors, they paid out US$11.03 per E-claim submitted and US$8.74 per paper claim submitted by providers, hence making a cost savings (benefits) of US$0.23 and US$0.97 for each electronic and paper claim submitted by providers respectively.

The average rejection rates for E-claims versus paper claims were 3% (0.001%–10%) versus 10% for the district hospitals, 1% versus 6 for regional hospitals, 0% versus 6% for tertiary hospital, 2% versus 10% NHIA, respectively. The highest rejection rate was seen among paper processing of claims for both providers and purchaser.

The average incremental benefits per E-claim were US$1.07 for district hospitals, US$1.07 for regional hospitals, US$−3.94 for tertiary hospital and US$−0.75 for CPC. Overall, the incremental benefit for all providers was estimated at US$10.72 and purchaser and providers was US$9.97 per E-claim.

### Incremental benefit–cost ratio of processing E-claims and paper claims
Table 4 presents a summary of the IBCRs by study population (study units).

The IBCR was 3.05 for district hospitals, −4.69 for regional hospitals, 12.72 for tertiary hospital and −8.28 for CPC. Two out of four district hospitals recorded IBCRs of less than one: St Martin's hospital (−6.65) and La general hospital (0.86). Three regional hospital also had an IBCR of less than one: Koforidua regional hospital (−5.28), Police hospital (−0.14) and Efia Nkwanta regional hospital (−81.49). The overall IBCR for all providers was −25.09 and that for the health system was −33.49.

Table 5 further presents the IBCR of all providers nationwide submitting claims to the NHIS CPC as at the end of 2014. The IBCR extrapolated for providers and purchaser nationwide were 3.05 for district hospitals, −4.69 for regional hospitals, 12.72 for tertiary hospitals and −8.28 for the CPC. IBCR for all providers nationwide was 11.08 and the entire health system was 2.79.

### Sensitivity analysis
The results of the sensitivity analysis showed that increasing the discount rate for both costs and benefits led to reductions in the IBCRs for each study unit and vice versa. However, this change did not influence the direction of the findings: from being efficient to not efficient and vice versa. For example, a 10% discount rate did not change the claims processing of a provider or purchaser from being profitable to unprofitable. When the proportions of E-claims to total claims were increased, the IBCR of the NHIA CPC reduced. For instance, increasing the proportion of E-claims from the base of 29%–50% as

**Table 3** Estimated benefits of E-claims and paper claims by providers and purchaser, 2014

| Study population | Value per claim submitted | | Rejection rate (%) | | Value per claim reimbursed (US$) (accounting for rejection rate) | | Incremental benefits (US$) |
|---|---|---|---|---|---|---|---|
| | E-claims | Paper claims | E-claims | Paper claims | E-claims | Paper claims | |
| Providers | | | | | | | |
| District | | | | | | | |
| Atebubu hospital | 9.06 | 8.82 | 10.00 | 20.00 | 8.16 | 7.05 | 1.10 |
| St. Martins hospital | 9.81 | 12.42 | 0.00 | 5.00 | 9.81 | 11.80 | (1.99) |
| Takoradi hospital | 9.07 | 4.65 | 0.10 | 0.10 | 9.06 | 4.64 | 4.42 |
| La general hospital | 10.15 | 9.67 | 0.00 | 3.00 | 10.15 | 9.38 | 0.77 |
| *Average district* | *9.52* | *8.89* | – | – | *9.29* | *8.22* | *1.07* |
| *Total district* | *38.08* | *35.55* | – | – | *37.17* | *32.87* | *4.29* |
| Regional | 0.00 | 0.00 | | | | | |
| Sunyani hospital | 15.94 | 14.51 | 2.40 | 3.00 | 15.56 | 14.08 | 1.48 |
| Koforidua hospital | 16.62 | 15.13 | – | 12.50 | 16.62 | 13.24 | 3.38 |
| Ridge hospital | 17.06 | 14.21 | 0.30 | 3.00 | 17.01 | 13.78 | 3.22 |
| Police hospital | 12.58 | 14.53 | 0.50 | 15.00 | 12.52 | 12.35 | 0.17 |
| Volta hospital | 12.98 | 16.44 | – | 0.00 | 12.98 | 16.44 | −3.46 |
| Effia Nkwanta hospital | 15.05 | 15.15 | 0.60 | 12.00 | 14.96 | 13.33 | 1.63 |
| *Average region* | *15.04* | *15.00* | – | – | *14.96* | *13.87* | *1.07* |
| *Total region* | *90.23* | *89.97* | – | – | *89.64* | *83.22* | *6.42* |
| Tertiary | | | | | | | |
| 37 military hospital | 14.31 | 19.48 | – | 6.30 | 14.31 | 18.26 | (3.94) |
| All providers | | | | | | | |
| Average | 11.66 | 11.41 | – | – | 11.55 | 10.50 | 0.97 |
| Total | 128.31 | 125.52 | – | – | 126.81 | 116.09 | 6.77 |
| Purchaser | | | | | | | |
| CPC (NHIA) | 11.26 | 9.71 | 2.00 | 10.00 | 0.23 | 0.97 | (0.75) |
| Health system (all providers and purchaser) | | | | | | | |
| Average | 11.63 | 11.27 | – | – | 10.59 | 9.76 | 0.83 |
| Total | 139.58 | 135.23 | – | – | 127.04 | 117.07 | 6.03 |

All costs were estimated in 2014. A conversion rate of GHC3.00 per US$1 was used.
CPC, claims processing centre; E-claims, electronic claims; NHIA, National Health Insurance Act.

against 50% paper claims (from 71% at base) reduced the IBCR from 15.14 to 2.32. The analysis shows that if all claims were processed electronically, 100% of the IBCR will be less than one (0.76), indicating that the incremental cost of processing additional claims exceeds incremental benefits. The CPC of NHIS can only achieve IBCR of one when 81% of claims are processed electronically and 19% processed manually. Any higher proportion of E-claims worsens the IBCR.

## DISCUSSION

Analysis of the benefits and costs of the E-claims and paper claims processing systems of the NHIA show that electronic processing of claims is more efficient than the manual system. On the average, all providers spent less cost in processing E-claims compared with paper claims. However, contrary to what was expected, the E-claims was more labour intensive. This could be attributed to providers associating efficiency of claims processing to the number of staff assigned to it: the higher the number of staff, the more efficient the processing and vice versa. Therefore, a more cost saving approach in terms of less labour under the E-claims could improve the efficiency gains by reducing the costs of processing. On the other hand, the purchaser (NHIA) spent more money to process claims electronically compared with manual system. The

**Table 4** Incremental benefit–cost ratio of processing E-claims and paper claims by healthcare providers and purchaser, 2014

| Study population | Incremental benefit–cost ratios |
|---|---|
| Providers | |
| District | |
| Atebubu hospital | 22.07 |
| St. Martins hospital | (6.65) |
| Takoradi hospital | 25.97 |
| La general hospital | 0.86 |
| *Average district* | *3.05* |
| *Total district* | *3.05* |
| Regional | |
| Sunyani hospital | 2.43 |
| Koforidua hospital | (5.28) |
| Ridge hospital | 6.58 |
| Police hospital | (0.14) |
| Volta hospital | 5.77 |
| Effia nkwanta hospital | (81.49) |
| *Average region* | *(4.69)* |
| *Total region* | *(4.69)* |
| Tertiary | |
| 37 military hospital | 12.72 |
| All providers | |
| Average | (48.71) |
| Total | (25.09) |
| Purchaser | |
| CPC (NHIA) | (8.28) |
| Health system (all providers and purchaser) | |
| Average | (83.09) |
| Total | (33.49) |

All costs were estimated in 2014. A conversion rate of GHC3.00 per US$1 was used.
Values in parenthesis denote negative values.
CPC, claims processing centre; E-claims, electronic claims; NHIA, National Health Insurance Act.

**Table 5** Extrapolated IBCR for all providers submitting claims to the NHIA CPC as at the end of year 2014

| Study population | Incremental cost (US$) | Incremental benefits (US$) | IBCR |
|---|---|---|---|
| Providers* | | | |
| All district hospitals | 347.98 | 1706.80 | 4.90 |
| All regional hospitals | (231.64) | 764.98 | (3.30) |
| Tertiary hospitals | (307.18) | (3953.13) | 12.87 |
| *All providers* | *(63.61)* | ***2239.18*** | *(35.20)* |
| Purchaser | | | |
| CPC | 89.97 | 2300.02 | 25.56 |
| Health system | | | |
| All providers and purchaser | 26.36 | 2373.95 | 90.06 |
| Providers† | | | |
| All district hospitals | 116.33 | 354.31 | 3.05 |
| All regional hospitals | (1.83) | 8.56 | (4.69) |
| Tertiary hospitals | (0.31) | (3.94) | 12.72 |
| All providers | | | |
| Average | (91.53) | 3633.04 | (39.69) |
| *Total* | *114.19* | *358.93* | *11.08* |
| Purchaser | | | |
| CPC | 0.09 | (0.75) | (8.28) |
| Health system (all providers and purchaser) | | | |
| Average | (3.4) | 282.52 | (83.09) |
| Total | *114.28* | *358.18* | *2.79* |

*Providers submitting claims to CPC by end 2014; all costs and benefits were estimated in 2014. A conversion rate of GHC3.00 per US$1 was used; values in parenthesis denote negative values.
†All providers nationwide; all costs and benefits were estimated in 2014; a conversion rate of GHC3.00 per US$1 was used; values in parenthesis denote negative values; as at the period of data collection, there were 330 districts hospitals, eight regional hospitals, one tertiary hospital and one NHIS CPC using E-claims processing.
CPC, claims processing centre; IBCR, incremental benefit–cost ratios; NHIA, National Health Insurance Act; NHIS, National Health Insurance Scheme.

reason for this is not readily known but might be due to maintenance cost of the E-claims infrastructure. The E-claims set-up requires periodic repairs and upgrade by the service provide to make it more robust and efficient, and this comes with cost to the NHIA.

Generally, the healthcare providers and the purchaser (NHIA) benefit more from processing claims using the electronic system than manual one. This assertion stems from the higher rejection rates from the E-claims processing system, compared with the paper claims processing system: 7.26% versus 1.26% for all providers and 10% versus 2% for the NHIS. Higher claims rejection by E-claims implies better ability of the purchaser to

detect errors in submitted claims, but it also motivates efficiency on the part of providers to complete claims forms better. These findings confirm the study by Nsiah-Boateng *et al*, which reported that the E-claims processing system has a higher claims rejection rate than the paper claims processing system, and concluded that the former could reduce cost to the NHIS. However, the claims rejection rates reported by our study were different from that

reported by Nsiah-Boateng *et al.*[9] For example, while they reported an adjustment rate of 17.9% for district hospitals using E-claims, we estimated 10% claims rejection rate for district hospitals. The difference in the rejection rate size could be attributed to the different methodological approaches employed to estimate effectiveness of E-claims processing system, the number and type of facilities used for both studies, and the study period.

This study also reveals a reduction in the volumes of claims submitted by providers under the E-claims compared with the paper claims processing system. This reduction was very apparent at the district and regional hospitals. The reasons for this reduction are not clearly known, but they could be attributed to healthcare providers' knowledge of the ability of the electronic claims processing to streamline claims and reduce the number of spurious claims submitted to the purchaser, NHIA.

Overall, an important implication of the findings of this study for policy is that the electronic processing of claims for reimbursement is efficient from the health system payer's (purchaser). Therefore, E-claims has the potential to reduce costs to the insurance system overall. In addition, E-claims improves claims processing times from both provider and insurance perspectives, with potential improvement in quality. Moreover, providers are making incremental gains from processing claims electronically compared with the manual system. The additional costs incurred from E-claims could be attributed to high start-up cost and labour cost. Therefore, it is anticipated that, when evaluated over a period of more than 1 year, these costs could reduce further.

It is important to note that for this study, benefits data used were claims rejection rates, which were applied on volumes and values of claims reimbursed to providers by the NHIA. Nevertheless, other important benefits of the E-claim system are worth mentioning. An essential benefit of E-claims over paper claim is that E-claims ensures that providers tend to rely on the stability of funds from the NHIS after providing services for clients. Furthermore, E-claims reduces the time between claims submission and reimbursement as reported by both NHIS staff and providers involved in this study. On the insurance side, there are savings on cost of operation and management of claims. The electronic system also provides a platform to identify actions of providers that lead to fraud in the system. On both the insurance (payer/purchaser) and provider sides, the electronic system improves the capacity to manage claims.

## Study limitations

The study findings notwithstanding, some limitations are worth mentioning. First, difficulties in quantifying some of the benefits of the E-claims into monetary terms, for example as discussed in the previous paragraph. Thus, methodological challenges with cost–benefit analysis hinder the use of some of the benefits in a cost–benefit analysis.[15–18] It is important to note the possibility of underestimation of the IBCRs reported in this study with the exclusion of such key benefits of electronic claims processing. With the advancement in the methods for efficiency analysis, it will be possible to measure and incorporate indicators such as fraud, predictability in funds and turnaround time in the estimation of the efficiency of electronic claims in the foreseeable future. Second, with Ghana's NHIS being one of first country-level social HIP in sub-Saharan Africa, there is little evidence elsewhere regarding the use of electronic claims processing that provides a comparison to Ghana, therefore, it is difficult to identify studies assessing the comparative benefits and costs of electronic versus paper claims processing for a national health insurance programme.

## CONCLUSION

The findings of this study show that the E-claims processing system is the economically preferred alternative to the paper claims. The study reinforces the recommendation of an earlier study on the scaling up of the E-claims system, even though for different reasons. Processing claims electronically improves the efficiency of the NHIS and reduces the number of claims rejected for payment on both the providers and the NHIA side, therefore, increasing their benefits compared with the manual processing of claims. However, to attain efficiency from the providers' side, the number of personnel assigned to E-claims processing would have to be reduced.

**Acknowledgements** Authors acknowledge the following for various contributions to the study: Lydia Dsane-Selby, Nathaniel Otoo, Rockson Atakole and Anthony Gingong.

**Contributors** JN and WH conceived the study. JN designed the study, collected and analysed data. JN and RA prepared the first draft of the manuscript. WH and ATS reviewed and provided substantial comments to first draft. All authors read and approved the final manuscript.

**Funding** This study was funded by the World Bank under its Health Insurance Project.

**Disclaimer** The views expressed in this article are those of the authors, and do not represent those of the funder or institutions of authors.

**Competing interests** JN received consultancy fees to undertake the study. ATS and WH are both employees of the funder.

**Patient and public involvement** Patients and/or the public were not involved in the design, or conduct, or reporting or dissemination plans of this research.

**Patient consent for publication** Not required.

**Ethics approval** Not applicable.

**Provenance and peer review** Not commissioned; externally peer reviewed.

**Data availability statement** Data are available upon reasonable request. The data set analysed during the current study are available from the corresponding author on reasonable request.

**ORCID iD**
Justice Nonvignon http://orcid.org/0000-0002-5265-2209

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
