## [Reviewer comments · BMJ Open]

ARTICLE DETAILS

TITLE (PROVISIONAL)	Benefit cost analysis of electronic claims processing under Ghana's National Health Insurance Scheme
AUTHORS	Nonvignon, Justice; Addo, Rebecca; Wang, Huihui; Seddoh, Anthony

VERSION 1 – REVIEW

REVIEWER	Ramadani, Royasia University of Indonesia, Center for Health Economics and Policy Studies
REVIEW RETURNED	08-Oct-2021

GENERAL COMMENTS	This is an excellent paper, and the finding has an important implication in informing the government and NHIA to expand the use of e-claim process. Though the benefit of using e-claims and have the national health insurance database is clear to provide more thorough assessment, monitoring and evaluation, this paper provide the calculation of the benefit using benefit cost analysis which presented in IBCR. This paper also added the information to the science community and to convince the government especially to moving on the UHC strengthening and moving to the e-claim database will provide the national health insurance database. Though this paper should be able to add more clarity, especially in the methods section, references and the implication of the research. The limitation of the study has to be articulated. See below attached my details comment in the documents 1. The intervention (piloting) should be explained in more details. what type of intervention in supporting the shifting from paper based and e-claim in 2014. This is to give a knowledge to the reader whether the intervention given to shift from e-claim based- will also likely to change the behaviour of provider which later affecting the outcome (here measured as outcome associated the benefit (percent of claim being reimbursed compared to rejected claim). And to make sure there is no spill over effect of the intervention2. The need to articulated costs information3. The need to articulated the sensitivity analysis, how many percent of the range discount rate being performed4. Most importantly, there is missing information on how author measured and valuing the Incurred But not Reported claim (IBNR) claim: here? Also how long the validity of the claim in the Ghana
--

	health insurance system? . There is no information how the IBNR is adjusted in the model 5. Also, what have been missing is that the information on what are the variables inputted in the e-claim, and how this database also in the future can be used to analysed the efficiency of the Ghana Health insurance system and many research, monev, can be conducted using e-claim database. This should be incorporated as well in one of the benefit from the health care system side (otherwise should be mentioned if its not calculated)
--	--

REVIEWER	Chola, Lumbwe Norwegian Institute of Public Health
REVIEW RETURNED	22-Dec-2021

GENERAL COMMENTS	Review comments Benefit cost analysis of electronic claims processing under Ghana's National Health Insurance Scheme General comments This is a very interesting but challenging analysis and I commend the authors on making this effort to add to the literature of an area where there is literally no information. Indeed as the authors point out, it is difficult to measure the costs and benefits of any records system. The authors will do well to highlight these issues comprehensively in the background section. Perhaps a conceptual framework of an electronic records processing system could be added – you will find a few of these in the literature that discuss the merits and demerits of the two systems. The potential issues such as time to processing, quality of service etc could be discussed. Further, the manuscript could benefit from a detailed discussion of the choice of cost and benefit parameters. This is brought up a few times in the discussion section, but could also be included in the methods section. Methods section Lines 19-24, page 4. Please add a citation to the discussion of BCA and its underpinnings Lines 39-46, page 4. Give more information about the population and sample. How many providers are in the system in total? What were the motivations for the purposively selected providers? How did you ensure representativeness etc. Lines 25-26, page 5. Provide a justification for using 3% discount rate Page 7 – add a rationale for choice of benefit and cost parameters Page 7 lines 47-50 – this information on the providers could have been added earlier to the section on study population. Results section Page 11 , line 15 – an interesting sentence about the rejection rate. Please explain why this is the case. Page 14 'sensitivity analysis section' the authors discuss show that the discount rate is an important factor. In view of this, is 3% an appropriate rate to use in the analysis?
---

	Discussion Lines 21-22, page 15. The authors should explain why the rejection rate is different between the two systems. Lines 52-60, page 15. Could you please provide a citation for this? Also, it could be good if some of this is discussed upfront in the background section as I suggested above. Maybe you need some kind of a conceptual framework that helps the reader visualize what the costs and benefits of the systems could be. General comment on the discussion: Please also bring out the clearly the research and policy implications of the findings. These are touched upon, but not very clearly outlined.
--	---

VERSION 1 – AUTHOR RESPONSE

Reviewer 1

Comment

This is an excellent paper, and the finding has an important implication in informing the government and NHIA to expand the use of e-claim process. Though the benefit of using e-claims and have the national health insurance database is clear to provide more thorough assessment, monitoring and evaluation, this paper provide the calculation of the benefit using benefit cost analysis which presented in IBCR

Response

Thank you

Comment

This paper also added the information to the science community and to convince the government especially to moving on the UHC strengthening and moving to the e-claim database will provide the national health insurance database

Response

Thank you

Comment

Though this paper should be able to add more clarity, especially in the methods section, references and the implication of the research. The limitation of the study has to be articulated. See below attached my details comment in the documents:

Response

These have been done

Comment

The intervention (piloting) should be explained in more details. what type of intervention in supporting the shifting from paper based and e-claim in 2014. This is to give a knowledge to the reader whether the intervention given to shift from e-claim based-will also likely to change the behaviour of provider which later affecting the outcome (here measured as outcome associated the benefit (percent of claim being reimbursed compared to rejected claim). And to make sure there is no spill over effect of the intervention

Response

We had a section in the methods but have now moved that to the background describing the E-claims system

Comment

The need to articulated costs information

Response

Please refer to sub-section ‘Estimating costs’ for cost parameters included in the analysis and a detailed description of how each cost centre was estimated. Table 1 also provides a summary of

formulas utilised in the cost estimation. Justification has now been provided for the choice of cost parameters.

Comment

The need to articulate the sensitivity analysis, how many percent of the range discount rate being performed

Response

The discount rates were varied within a range of 0% to 10%, excluding 3%, the base case. This has been added under the methods section; subsection: sensitivity analyses, page 9.

Comment

Most importantly, there is missing information on how author measured and valuing the Incurred But not Reported claim (IBNR) claim: here? Also how long the validity of the claim in the Ghana health insurance system? There is no information how the IBNR is adjusted in the model

Response

We did not measure IBNR claims in this study. We did factor. We did, however, factor in incurred but not reimbursed claims, which were factored in deriving benefits. In Ghana, claims (both paper and electronic claims) have a validity period of 90 days. However, the purchaser usually have some flexibility around this, if providers give tangible reasons why claims are submitted late, in which case they would need permission from the purchaser to submit late

Comment

Also, what have been missing is that the information on what are the variables inputted in the e-claim, and how this database also in the future can be used to analysed the efficiency of the Ghana Health insurance system and many research, money, can be conducted using e-claim database. This should be incorporated as well in one of the benefit from the health care system side (otherwise should be mentioned if its not calculated)

Response

The E-claims contains the same parameters as the manual claims, mainly clinical information on the treatment and key patient characteristics, as well as the costs of each bundle of service delivered to the patient. As measured in this study, in addition to these data, one would require other data relating to resources used in processing the claims, such as capital, human resource costs and productivity time, which are not recorded on the claims – details of these are available under Estimating costs section of the methods.

Reviewer 2

Comment

This is a very interesting but challenging analysis and I commend the authors on making this effort to add to the literature of an area where there is literally no information. Indeed as the authors point out, it is difficult to measure the costs and benefits of any records system:

Response

Thank you

Comment

The authors will do well to highlight these issues comprehensively in the background section. Perhaps a conceptual framework of an electronic records processing system could be added – you will find a few of these in the literature that discuss the merits and demerits of the two systems.

Response

The issues have been highlighted in the background. In addition, we have explained the benefits of E-claims and, together with Figure 1 and explanation in the methods, these highlight the benefits and costs of the E-claims and manual claims systems.

Comment

The potential issues such as time to processing, quality of service etc. could be discussed.

Response

This has been done in the discussion

Comment

Further, the manuscript could benefit from a detailed discussion of the choice of cost and benefit parameters. This is brought up a few times in the discussion section, but could also be included in the methods section.

Response

We have included this on pages 6 and 7

Methods section

Comment

Lines 19-24, page 4. Please add a citation to the discussion of BCA and its underpinnings

Response

We have provided citation in the methods

Comment

Lines 39-46, page 4. Give more information about the population and sample. How many providers are in the system in total? What were the motivations for the purposively selected providers? How did you ensure representativeness etc.

Response

We have included this information in the study population section of the manuscript. Out of 29 hospitals that implemented E-claims as at the time of the study, the purposive selection of 11 was based on how long they had submitted E-claims (we included those who had submitted E-claims for at least one year to allow us better capture the costs and benefits). In this study, we sought to capture the different levels of facilities (i.e. primary, secondary, tertiary) given the different scales of operation.

Comment

Lines 25-26, page 5. Provide a justification for using 3% discount rate

Response

Justification provided page 7

Comment

Page 7 – add a rationale for choice of benefit and cost parameters

Response

We have included this on pages 6 and 7

Comment

Page 7 lines 47-50 – this information on the providers could have been added earlier to the section on study population.

Response

Thank you. This has been moved to the study population section on page 5.

Results section

Comment

Page 11 , line 15 – an interesting sentence about the rejection rate. Please explain why this is the case.

Response

This is explained in paragraph 2 of the discussion

Comment

Page 14 'sensitivity analysis section' the authors discuss show that the discount rate is an important factor. In view of this, is 3% an appropriate rate to use in the analysis?

Response

Yes. We believe 3% is an appropriate rate for the analysis as this is the recommended rate to be used for economic evaluations in low and middle income countries (refer to page 7, under 'estimation costs' subsection for supporting references)

Discussion

Comment

Lines 21-22, page 15. The authors should explain why the rejection rate is different between the two systems.

Response

This has been done. Claims rejection reflects the ability of each system to detect errors in submitted claims. Therefore, the higher rejection by the E-claims implies its better ability to detect errors, which is one of the reasons the E-claims was set up, as reduction in payout of erroneous claims leads to reduction in overall costs to the purchaser.

Comment

Lines 52-60, page 15. Could you please provide a citation for this? Also, it could be good if some of this is discussed upfront in the background section as I suggested above. Maybe you need some kind of a conceptual framework that helps the reader visualize what the costs and benefits of the systems could be.

Response

Done

General comment on discussion

Comment

Please also bring out the clearly the research and policy implications of the findings. These are touched upon, but not very clearly outlined.

Response

Done

VERSION 2 – REVIEW

REVIEWER	Ramadani, Royasia University of Indonesia, Center for Health Economics and Policy Studies
REVIEW RETURNED	01-Apr-2022

GENERAL COMMENTS	The authors made a clear revision. There is a minor revision in the reference and wording template need to address. Tables 2 and 3 are good to add the number of incremental costs (table 2) and incremental benefits (table 4) and present Table 4 in different ways. In the discussion, it was mentioned that one of the benefits of moving to e claim for purchasers is a higher rejection rate. Does moving to e-claim will affect hospital cash flow or th rejection of the claim is associated to moral hazards? Please discuss
--

REVIEWER	Chola, Lumbwe Norwegian Institute of Public Health
REVIEW RETURNED	13-Mar-2022

GENERAL COMMENTS	None. All comments are thoroughly addressed and the manuscript is clear.
--

VERSION 2 – AUTHOR RESPONSE

Reviewer: 1

Ms. Royasia Ramadani, University of Indonesia

Comments to the Author:

The authors made a clear revision. There is a minor revision in the reference and wording template need to address. Tables 2 and 3 are good to add the number of incremental costs (table 2) and incremental benefits (table 4) and present Table 4 in different ways. In the discussion, it was mentioned that one of the benefits of moving to e claim for purchasers is a higher rejection rate. Does moving to e-claim will affect hospital cash flow or th rejection of the claim is associated to moral hazards? Please discuss

Response: The manuscript has been revised as suggested by the reviewer